# Decision-Focused Surrogate Modeling for Mixed-Integer Linear Optimization

**Shivi Dixit**                                              *dixit064@umn.edu*
*Department of Chemical Engineering and Materials Science,*
*University of Minnesota Twin Cities, MN, USA.*

**Rishabh Gupta**                                    *rishabh.gupta1@exxonmobil.com*
*ExxonMobil Technology and Engineering Company, Spring, TX, USA.*

**Qi Zhang**                                                   *qizh@umn.edu*
*Department of Chemical Engineering and Materials Science,*
*University of Minnesota Twin Cities, MN, USA.*

**Reviewed on OpenReview:** *https://openreview.net/forum?id=A6tOXkkE4Z*

## Abstract

Mixed-integer optimization is at the core of many online decision-making systems that demand frequent updates of decisions in real time. However, due to their combinatorial nature, mixed-integer linear programs (MILPs) can be difficult to solve, rendering them often unsuitable for time-critical online applications. To address this challenge, we develop a data-driven approach for constructing surrogate optimization models in the form of linear programs (LPs) that can be solved much more efficiently than the corresponding MILPs. We train these surrogate LPs in a decision-focused manner such that for different model inputs, they achieve the same or close to the same optimal solutions as the original MILPs. One key advantage of the proposed method is that it allows the incorporation of all of the original MILP's linear constraints, which significantly increases the likelihood of obtaining feasible predicted solutions. Results from two computational case studies indicate that this decision-focused surrogate modeling approach is highly data-efficient and provides very accurate predictions of the optimal solutions. In these examples, the resulting surrogate LPs outperform state-of-the-art neural-network-based optimization proxies.

## 1 Introduction

Effectively operating complex systems in online environments, where input parameters are constantly changing, requires efficient decision making in real time. Many online decision-making frameworks involve the solving of mathematical optimization problems; however, often the computational complexity of the given optimization problem presents a major challenge such that the long solution time renders it ineffective in real-time applications. A common approach to tackling this challenge is to perform the online optimization with a *surrogate model*, which is an approximation of the original model that can be solved more efficiently (Bhosekar & Ierapetritou, 2018).

Surrogate modeling for optimization typically involves three steps: (i) identify the complicating constraints in the original optimization model, (ii) generate surrogate models for the complicating constraint functions, and (iii) replace the complicating constraints with the surrogates to obtain a surrogate optimization model that can be solved more efficiently. This approach is especially effective in applications where the computational complexity of the problem is only due to a small part of the model such that one can keep most of the original constraints. Here, a major underlying assumption is that a surrogate model that provides a good approximation of the complicating constraints will also, once incorporated into the optimization problem,

lead to solutions that are close to the true optimal solutions. However, this assumption often does not hold. The original optimization problem and the surrogate optimization problem may achieve very different optimal solutions despite having a highly accurate (but not perfect) embedded surrogate model.

Recently, Gupta & Zhang (2024) have developed a new surrogate modeling framework that explicitly aims to construct surrogate models that minimize the *decision prediction error* defined as the difference between the optimal solutions of the original and the surrogate optimization problems; it is hence referred to as *decision-focused surrogate modeling* (DFSM). DFSM has been applied to construct convex *decision-focused surrogate optimization models* (DFSOMs) for nonconvex nonlinear programs that are difficult to solve to global optimality Gupta & Zhang (2024); however, these problems only involve continuous variables. In this work, we extend the DFSM framework to MILPs, which arise in many real-time decision-making problems such as online production scheduling and hybrid model predictive control. Due to their combinatorial nature, MILPs can take long times to solve. Here, we propose to construct DFSOMs in the form of LPs that are computationally significantly more efficient and achieve optimal solutions close to the ones of the original MILPs.

The remainder of this paper is organized as follows. We first provide a review of related works in Section 2, focusing on existing data-driven and parametric methods for solving online mixed-integer optimization problems. The proposed DFSM approach for MILPs is presented in Section 3, while its utility is highlighted in two computational case studies in Section 4. Finally, in Section 5, we close with some concluding remarks.

## 2 Related work

In this section, we provide a brief review of related works. Note that we do not review the vast literature on surrogate modeling as the methods described therein are of less relevance to us given our specific focus on MILPs and decision-focused modeling approaches.

### 2.1 ML-based optimization proxies

The line of research that is most closely related to this work is the one on constructing optimization proxies using machine learning (ML). Here, the goal is to learn a direct mapping of the input parameters of an optimization problem to its optimal solution. In that sense, it has the same objective as DFSM, namely to obtain a model that predicts optimal solutions close to the true ones. It uses the same offline generated data consisting of different model inputs and the corresponding true optimal solutions. The main difference is that while DFSM trains a model in the form of a surrogate optimization problem, optimization proxies are ML models, most commonly neural networks (NNs) (Sun et al., 2018; Pan et al., 2019; Karg & Lucia, 2020; Kumar et al., 2021).

Optimization proxies take advantage of the expressive power of deep NNs; however, in deep learning, it is difficult to enforce constraints on the predictions, which often leads to predicted solutions that are infeasible in the original optimization problem. In their work on constructing NN-based optimization proxies for the AC optimal power flow (OPF) problem, Zamzam & Baker (2020) ensure feasibility by generating a training set of strictly feasible solutions through a modified AC OPF formulation and by using the natural bounds of the sigmoid activation function to enforce generation and voltage limits. When also addressing the AC OPF problem, Fioretto et al. (2020) consider constraints in their deep learning framework by augmenting the loss function with penalty terms derived from the Lagrangian dual of the original optimization problem and applying a subgradient method to update the Lagrange multipliers. This approach has also been applied to the security-constrained OPF problem (Velloso & Van Hentenryck, 2021) and job shop scheduling (Kotary et al., 2022). Most of these and similar approaches increase the likelihood of obtaining feasible predictions but cannot guarantee it; hence, in most cases, a feasibility restoration step is added to obtain a feasible solution. One common approach to correcting an infeasible prediction is to project it onto a suitable set such that the projection is a feasible solution (Chen et al., 2018; Paulson & Mesbah, 2020).

## 2.2 Decision-focused learning

We now describe a framework that is closely related to the DFSM approach but is not motivated by the need for fast online optimization. Instead, it addresses the following problem: In traditional data-driven optimization, we often follow a two-stage predict-then-optimize approach, i.e. we first predict unknown input parameters from data with external features and then solve the optimization problem with those predicted inputs. Here, the learning step focuses on minimizing the parameter estimation error; however, this does not necessarily lead to the best decisions (evaluated with the true parameter values) in the optimization step. In contrast, *decision-focused learning* (Wilder et al., 2019), also known as *smart predict-then-optimize* (Elmachtoub & Grigas, 2022) or *predict-and-optimize* (Mandi et al., 2020), integrates the two steps to explicitly account for the quality of the optimization solution in the learning of the model parameters.

Elmachtoub & Grigas (2022) consider decision-focused learning for optimization problems with a linear objective function and a closed convex feasible region. They develop a convex surrogate loss function that is Fisher consistent with respect to the original loss function and allows the use of stochastic gradient descent to efficiently solve the problem. The same algorithm has also been applied to combinatorial problems (Mandi et al., 2020). Decision-focused learning has further motivated research on differentiable optimization in deep learning (Amos & Kolter, 2017), which allows NNs to comprise differentiable layers that represent full optimization problems. Using this approach, quadratic programs (Donti et al., 2017), linear programs (Wilder et al., 2019), and mixed-integer linear programs (Ferber et al., 2020) have been considered in the training of decision-focused NNs. This approach is applied by Ferber et al. (2023) to address a problem of similar nature as our DFSM problem, namely to learn linear surrogate objective functions for discrete optimization problems with linear constraints but nonlinear objective functions.

## 2.3 ML-enhanced mixed-integer optimization

There is a rapidly growing body of literature on using ML to speed up the solution of mixed-integer optimization problems (Bengio et al., 2021). Our work as well as the ML-based optimization proxies reviewed in Section 2.1 fall into this broad category. Other strategies include using ML to learn more effective primal heuristics (Bengio et al., 2020; Shen et al., 2021), branching rules (Lodi & Zarpellon, 2017; Khalil et al., 2016), the set of active constraints at the optimal solution (Bertsimas & Stellato, 2022), and how to warm-start MILPs (Jiménez-Cordero et al., 2022). More closely related to our work are the contributions on learning how to improve the generation of cutting planes (Deza & Khalil, 2023; Tang et al., 2020; Huang et al., 2022). However, while these methods add cuts as part of a branch-and-cut algorithm, our approach learns parametric cuts that can be added immediately to the LP relaxation for a given model input, as discussed in more detail in Section 3.

## 2.4 Multiparametric programming

While DFSM and optimization proxies can generally only provide approximate optimal solutions, multiparametric programming is an exact approach that maps model inputs directly to the true optimal solutions through explicit functions (Oberdieck et al., 2016). The solution of a multiparametric LP follows from the basic sensitivity theorem (Fiacco & Ishizuka, 1990), which states that the active set remains unchanged in the neighborhood of the given vector of input parameters and the corresponding optimal solution. This results in the partitioning of the feasible parameter space into polytopic regions called critical regions. The goal of multiparametric programming is to obtain these critical regions and their corresponding optimal policies offline, then the online optimization reduces to selecting and applying the right policy for a given parameter vector that lies in the corresponding critical region.

When solving multiparametric MILPs, the same concept can be applied. One naive way is to enumerate all the possible discrete solutions and then treat the multiparametric MILP as a set of many multiparametric LPs (Roodman, 1972), one for each discrete solution. Thus, the resulting solution to the multiparametric MILP is again achieved by partitioning the parameter space into critical regions, each of which is associated with an optimal policy as an affine function of the model parameters. More advanced methods apply algorithms

based on branch-and-bound (Ohtake & Nishida, 1985; Acevedo & Pistikopoulos, 1999) and decomposition (Dua & Pistikopoulos, 2000).

Multiparametric programming has found several successful applications in explicit model predictive control (Alessio & Bemporad, 2009; Pistikopoulos et al., 2015) with extensions to other frameworks such as integrated design and operations (Burnak et al., 2019) and optimization under uncertainty (Charitopoulos et al., 2018). However, as the problem size grows with the number of decision variables, constraints, and model parameters, the construction of the full multiparametric solution becomes challenging as the number of critical regions grows exponentially with the problem size. Moreover, given a parameter vector, finding the critical region for that point becomes a point location problem, which can also be a computationally difficult problem for larger systems with many critical regions.

## 3 DFSM for MILPs

We consider MILPs, which we aim to solve efficiently in an online setting, given in the following general form:

$$
\begin{aligned}
\underset{x}{\text{minimize}} \quad & c(u)^\top x \\
& A(u)x \le b(u) \\
& x \in \mathbb{R}^m \times \mathbb{Z}^{n-m},
\end{aligned}
\tag{1}
$$

where $x$ is a vector of continuous and discrete variables. The cost vector, constraint matrix, and right-hand-side vector are denoted by $c(u)$, $A(u)$, and $b(u)$, respectively, which generally all depend on the input parameters $u$.

In general, the complicating constraints in an MILP are the integrality constraints on the discrete variables. Here, the key idea is to replace them with simple linear inequalities. This is motivated by a fundamental property of MILPs that can be stated as follows (Schrijver, 1998): Given a mixed-integer polyhedral set $\mathcal{S} = \{x \in \mathbb{R}^m \times \mathbb{Z}^{n-m} : Ax \le b\}$, there exist $\bar{A}$ and $\bar{b}$ such that the convex hull of $\mathcal{S}$ is $\text{conv}(\mathcal{S}) = \{x \in \mathbb{R}^n : Ax \le b, \bar{A}x \le \bar{b}\}$, as illustrated in Figure 1. Then, loosely speaking, an MILP with a cost vector $c$ and $\mathcal{S}$ as its feasible region will have the same optimal value as an LP with the same cost vector and constrained by $Ax \le b$ and $\bar{A}x \le \bar{b}$. Moreover, if that MILP has a unique optimal solution, the corresponding LP will have the same optimal solution, i.e. $\arg\min_x\{c^\top x : x \in \mathcal{S}\} = \arg\min_x\{c^\top x : x \in \text{conv}(\mathcal{S})\}$.

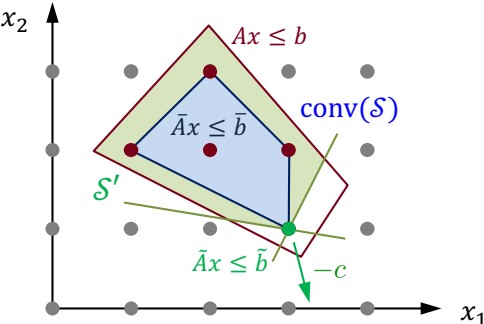

Figure 1: Illustrative example in which for the given cost vector $c$, the MILP with integer variables $x_1$ and $x_2$ and constraints $Ax \le b$ has the same optimal solution as the LPs with $\text{conv}(\mathcal{S})$ and $\mathcal{S}'$ as their feasible regions.

One does not have to recover the full convex hull to obtain an LP that achieves the same optimal solution as the MILP. As illustrated in Figure 1, for the given cost vector $c$, the LP with $\mathcal{S}' = \{x \in \mathbb{R}^n : Ax \le b, \tilde{A}x \le \tilde{b}\}$ as its feasible region, where $\tilde{A}x \le \tilde{b}$ are a smaller set of constraints, will equally lead to the same optimal solution. This idea forms the basis for traditional exact cutting-plane approaches, where linear inequalities are successively added to the linear relaxation of the MILP until the optimal solution is found (Marchand

et al., 2002). The downside of a cutting-plane algorithm (or one implemented in a branch-and-cut scheme) is that for every new instance of the MILP, new cutting planes need to be generated in the same iterative fashion, which can be too time-intensive for real-time applications.

In our DFSM approach, we propose to learn LPs with added linear inequalities parameterized by the input parameters $u$ as surrogate optimization models for the corresponding MILPs. As such, this approach can be interpreted as learning parametric cutting planes that are generated a priori and change automatically with each new instance given by new values for the input parameters $u$.

## 3.1 General formulation

Given an MILP of the form (1), the surrogate LP that we aim to construct can be written as follows:

$$\underset{x \in \mathbb{R}^n}{\text{minimize}} \quad c(u)^\top x \tag{2a}$$

$$A(u)x \le b(u) \tag{2b}$$

$$\tilde{A}(u, \theta)x \le \tilde{b}(u, \theta), \tag{2c}$$

where (2b) are the original inequality constraints of problem (1), and (2c) are the new set of linear constraints that we want to learn. The elements of both the constraint matrix $\tilde{A}$ and right-hand-side vector $\tilde{b}$ are functions of the input parameters $u$. These functional relationships are parameterized by the parameters $\theta$ and can in principle be of arbitrary complexity. In this work, we use low-order polynomials, which have proven to be sufficiently accurate in our case studies and also provide some computational advantages (see Section 3.2). In this case, $\theta$ are the coefficients of the polynomial terms. In Appendix A.1, we present an illustrative example that highlights how such parametric inequalities can help the surrogate LP recover the optimal solutions of the original MILP.

The DFSM problem for learning a DFSOM of the form (2) can be formulated as follows:

$$\underset{\theta \in \Theta, \, \tilde{x}}{\text{minimize}} \quad \sum_{i \in \mathcal{I}} \|x_i^* - \tilde{x}_i\|_1 \tag{3a}$$

$$\tilde{x}_i \in \underset{x \in \mathbb{R}^n}{\arg\min}\{c^\top x : A(\bar{u}_i)x \le b(\bar{u}_i), \, \tilde{A}(\bar{u}_i, \theta)x \le \tilde{b}(\bar{u}_i, \theta)\} \quad \forall \, i \in \mathcal{I}, \tag{3b}$$

where $\mathcal{I} = \{(\bar{u}_i, x_i^*)\}_{i=1}^N$ constitutes the training dataset with $x_i^*$ being the optimal solution to the original MILP (1) for input $\bar{u}_i$. As shown in (3a), we use the $\ell_1$-norm to measure the decision error, and the objective is to choose $\theta$ from some appropriate set $\Theta$ such that the sum of decision errors across all training data points is minimized. Problem (3) is a bilevel optimization problem with $N$ lower-level problems. As shown in (3b), each lower-level problem represents the surrogate LP for a particular input $\bar{u}_i$, and $\tilde{x}_i$ is constrained to be an optimal solution to that LP.

## 3.2 Solution strategy

To solve problem (3), we first reformulate it into a single-level optimization problem by replacing the lower-level problems with their KKT optimality conditions, which results in the following formulation:

$$\underset{\theta \in \Theta, \, \tilde{x}, \lambda, \mu}{\text{minimize}} \quad \sum_{i \in \mathcal{I}} \|x_i^* - \tilde{x}_i\|_1 \tag{4a}$$

$$c + A(\bar{u}_i)^\top \lambda_i + \tilde{A}(\bar{u}_i, \theta)^\top \mu_i = 0 \quad \forall \, i \in \mathcal{I} \tag{4b}$$

$$A(\bar{u}_i)\tilde{x}_i - b(\bar{u}_i) \le 0 \quad \forall \, i \in \mathcal{I} \tag{4c}$$

$$\tilde{A}(\bar{u}_i, \theta)\tilde{x}_i - \tilde{b}(\bar{u}_i, \theta) \le 0 \quad \forall \, i \in \mathcal{I} \tag{4d}$$

$$D(\lambda_i)(A(\bar{u}_i)\tilde{x}_i - b(\bar{u}_i)) = 0 \quad \forall \, i \in \mathcal{I} \tag{4e}$$

$$D(\mu_i)(\tilde{A}(\bar{u}_i, \theta)\tilde{x}_i - \tilde{b}(\bar{u}_i, \theta)) = 0 \quad \forall \, i \in \mathcal{I} \tag{4f}$$

$$\lambda_{ir} \ge 0, \, \mu_{ij} \ge 0, \, \tilde{x}_i \in \mathbb{R}^n \quad \forall \, i \in \mathcal{I}, \, r \in \mathcal{R}, \, j \in \mathcal{V}, \tag{4g}$$

where (4b) are the stationarity conditions, (4c)-(4d) are the primal feasibility conditions, (4e)-(4f) are the complementary slackness conditions, and (4g) are the dual feasibility conditions. The dual variables are denoted by $\lambda$ and $\mu$. In (4e)-(4f), $D(\cdot)$ denotes a diagonal matrix formed from a given vector, $\mathcal{R}$ represents the set of original constraints in the MILP, and $\mathcal{V}$ represents the set of additional constraints in the DFSOM.

Problem (4) is a nonconvex NLP that violates constraint qualification (due to, for example, the complementary slackness conditions), which means that standard NLP solvers cannot guarantee convergence to the optimal solution when directly applied to this problem. Here, we adopt the solution strategy proposed by Gupta & Zhang (2022) for data-driven inverse optimization problems, which turn out to have the same structure as the DFSM problem. We apply an exact penalty reformulation that is achieved by penalizing the violation of constraints (4b)-(4f) in the objective function using the $\ell_1$-norm. We can then iteratively update the penalty parameter until no more constraint violation is observed at which point the algorithm is guaranteed to have converged to a local solution of problem (4). The penalty reformulation takes the form of a multiconvex optimization problem in terms of the three sets of variables $\theta$, $\tilde{x}$, and $(\lambda, \mu)$, i.e. the problem is convex with respect to each of these three sets of variables. This structure can be exploited using a block coordinate descent (BCD) algorithm that allows an efficient decomposition of the problem, which has been crucial in solving large instances of the DFSM problem. For more details on the solution algorithm, see Appendix A.2.

## 4 Computational case studies

We conduct two computational case studies to assess the efficacy of the proposed DFSM approach. Both examples are representative of common real-time optimization problems involving both continuous and discrete decisions. All optimization problems were implemented in Julia v1.7.2 using the modeling environment JuMP v0.22.3 (Dunning et al., 2017). We applied Gurobi v10.3 to solve all LPs and MILPs, and all NLPs were solved using IPOPT v0.9.1. All DFSM instances were solved utilizing 24 cores and 60 GB memory on the Mesabi cluster of the Minnesota Supercomputing Institute (MSI) equipped with Intel Haswell E5-2680v3 processors. The codes developed for these case studies are available at `https://github.com/ddolab/DFSM-for-MILPs`.

### 4.1 Hybrid vehicle control

We first consider a hybrid vehicle control problem adapted from Takapoui et al. (2015). Given a hybrid vehicle with a battery, an electric generator, and an engine, the objective is to minimize the fuel consumption cost over a given time horizon while also maintaining a high terminal battery level. This problem can be formulated as the following MILP:

$$\underset{E, P^{\text{batt}}, P^{\text{eng}}, z}{\text{minimize}} \quad \sum_{t=0}^{T-1} (\alpha_t P_t^{\text{eng}} + \beta z_t) + \eta(E^{\max} - E_T) \tag{5a}$$

$$E_0 = E_{\text{init}} \tag{5b}$$

$$0 \le E_t \le E^{\max} \quad \forall t = 0, ...., T \tag{5c}$$

$$E_{t+1} = E_t - \tau P_t^{\text{batt}} \quad \forall t = 0, ...., T-1 \tag{5d}$$

$$0 \le P_t^{\text{eng}} \le z_t P^{\max}/S \quad \forall t = 0, ...., T-1 \tag{5e}$$

$$P_t^{\text{batt}} + P_t^{\text{eng}} \ge D_t \quad \forall t = 0, ...., T-1 \tag{5f}$$

$$z_t \in \{0, 1, .., S\} \quad \forall t = 0, ...., T-1. \tag{5g}$$

Here, the decision variables are given for each time period $t$, starting with the energy status of the battery $E_t$, then the power to or from the battery $P_t^{\text{batt}}$, the power from the engine $P_t^{\text{eng}}$, and the engine switch status $z_t$. The input parameters are the power demand $D_t$ over the $T$ time periods and the initial battery state $E_{\text{init}}$. The cost function to be minimized, (5a), contains the fuel cost in each time period given by $\alpha_t P_t + \beta_t z_t$ and a penalty on the deviation of the terminal battery charge from its maximum given by $\eta(E^{\max} - E_T)$. The initial battery state is given by (5b). The bounds on the battery state are stated in (5c). The battery charge balance is given by (5d), where $\tau$ denotes the length of each time interval. We assume that the engine can operate at different levels, modeled using the integer variable $z_t$. The different levels represent different

fractions of the maximum engine power available for use in a given time period. As per constraints (5e), when $z_t = 0$, no power can be derived from the engine, and when $z_t = S$, the maximum amount of power can be drawn. Finally, the combined power output from the battery and the engine must at least match the demand $D_t$ as stated in (5f).

### 4.1.1 Data generation and surrogate design

To generate different problem instances, we choose $\eta \sim \mathcal{N}(2.5, 5.5)$, $\alpha_t \sim \mathcal{N}(6, 16)$, $\beta_t \sim \mathcal{N}(0.5, 2)$, and $E_{\text{init}} \sim \mathcal{N}(90, 97)$, and set $\tau = 5$, $P^{\max} = 1$, and $E^{\max} = 100$. Moreover, the value of $S$ is also varied between $\{1, 2, 3\}$ across these instances. This way we generate 10 different instances, each representing a hybrid vehicle with different attributes. For each instance, the training dataset is generated by varying the power demand profile $D$, which is the model input, and solving the original MILP for that demand profile to obtain the corresponding optimal solution $(E^*, P^{\text{eng}*}, z^*)$. Repeating this process provides a set of data points, each consisting of an input-solution pair.

The additional linear inequalities in the postulated surrogate LP take the form of $\tilde{A}(D, \theta)[E \; P^{\text{eng}} \; z]^\top \leq \tilde{b}(D, \theta)$, where the number of constraints $|\mathcal{V}|$ is a hyperparameter that can be adjusted. Each element of $\tilde{A}$ and $\tilde{b}$ is a cubic polynomial in $D$ with $\theta$ being the coefficients. In this case, the time required to train the surrogate model for 100 training data points is around 10,000 seconds. This surrogate design remains the same across the 10 different instances; however, for each instance, different linear inequality parameters $\theta$ have to be derived.

We also compare the performance of the DFSM approach with that of three NN-based optimization proxies, one that applies a simple feedforward NN architecture, a second one where the loss function contains penalty terms derived from the Lagrangian dual of the optimization problem to impose feasibility on the predicted solutions (Fioretto et al., 2020), and a third one that is based on a graph NN (GNN) architecture. To represent the MILP as a GNN, we derive a bipartite graph whose two sets of nodes correspond to the problem's variables and constraints, respectively. An edge connects a variable node with a constraint node if that variable appears in the constraint. Also, features such as variable bounds and right-hand-side values are assigned to the nodes to fully describe the MILP. These NN-based optimization proxies are similarly trained in a decision-focused manner using the same datasets. See Appendix A.3 for more details on the design and training of the NN-based optimization proxies.

### 4.1.2 Surrogate model performance

All reported values are averaged over the 10 different randomly generated instances. For the DFSOM and the NN-based optimization proxies, if a predicted solution is not directly integer-feasible, an inexpensive projection-based feasibility restoration problem is solved to recover a feasible solution. More details on the feasibility restoration step can be found in Appendix A.4.

The decision prediction errors and optimality gaps for each instance are determined using 100 unseen test data points, each given by a different power demand profile. The relative prediction error is defined as the $\ell_1$-norm of the difference between the predicted solution (after feasibility restoration) and true optimal solution divided by the $\ell_1$-norm of the true optimal solution. Similarly, the optimality gap is the difference between the costs of the predicted and true optimal solutions. The size of the problem increases with $T$. In the following, we present results for the case with $T = 30$, which has 61 continuous variables, 30 integer variables, and 122 constraints in the resulting MILP; additional results for $T = 10$ and $T = 20$ are shown in Appendix A.5.

Figure 2 compares the performance of the DFSOM and the NN-based optimization proxies. For these methods, it shows how the prediction errors and optimality gaps change with the number of training data points. Note that while the maximum number of training data points we use to construct each DFSOM is 100, we use up to 500 data points to train each of the three NN-based optimization proxies. Also, to aid the interpretation of the results, we show the prediction errors with respect to the discrete (Figure 2(a)) and continuous (Figure 2(b)) decisions separately. We observe that the Lagrangian-based NN (NN+LD) achieves slightly better solutions than the standard NN, likely because its predictions are closer to being integer-feasible. The GNN performs significantly better. Yet the DFSOM outperforms all three NN-based optimization proxies in terms

of prediction accuracy, where the prediction errors plateau at much lower values as the size of the training dataset increases. DFSM also proves to be very data-efficient as it achieves a high prediction accuracy with only 50 data points while the NNs seem to require much more data.

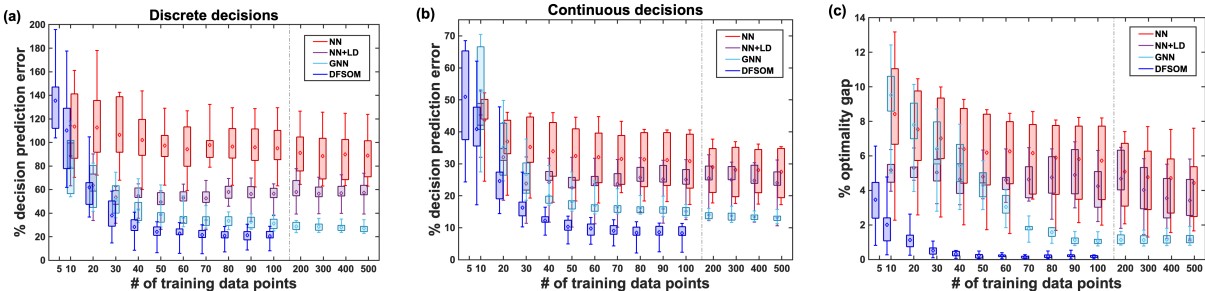

Figure 2: DFSOM performance compared to NN-based optimization proxies in the hybrid vehicle control problem.

Notice that the relative prediction errors that the DFSOM achieves are about 20% and 8% in terms of the discrete and continuous decisions, respectively, which seem relatively high. The corresponding optimality gaps, however, are close to zero, as shown in Figure 2(c). While surprising at first, upon closer investigation, we can attribute this behavior to the specific nature of the hybrid vehicle control problem. Given a fixed solution in terms of the discrete variables $z$ for problem (5), due to the flexibility that the battery provides, there are many solutions in terms of the continuous variables that are optimal or close to optimal. Moreover, as the cost parameter $\alpha_t$ has a much larger value than $\beta_t$, the contribution of the continuous variables to the total cost outweighs that of the discrete variables. As a result, from a cost standpoint, it is more important to predict the continuous variables accurately, hence the difference in the relative prediction errors.

We see that it is important to predict integer-feasible solutions, which is where DFSM has an advantage as it keeps all linear inequalities from the original MILP in the DFSOM. Here, the difference between the solutions directly predicted by the DFSOM and the integer-feasible solutions obtained after feasibility restoration is on average 6.6%. The standard NN tends to predict solutions that are far from being integer-feasible such that the obtained solutions after feasibility restoration are highly suboptimal, as indicated in Figure 2(c). This is also reflected in the on average 26.8% difference between the solutions predicted by the NN and those post feasibility restoration. This difference decreases to 23.1% in the case of the Lagrangian-based NN, which is still much higher than what is achieved by the DFSOM. In the case of the GNN, the difference further decreases to 12.8%, highlighting the advantage of using GNNs as NN-based optimization proxies for MILPs in problems of this format.

We further investigate the impact of the number of added constraints, $|\mathcal{V}|$, on the performance of the resulting DFSOM. Figure 3 shows the decision prediction errors and optimality gaps for different $|\mathcal{V}|$. Here, we can see that when $T/3$ linear inequalities are added, the DFSOM cannot reduce the prediction error to a reasonable level even with a large number of training data points. As we increase $|\mathcal{V}|$ to $T$ and then to $2T$, the prediction accuracy increases significantly. However, there is only a marginal improvement from $|\mathcal{V}| = 2T$ to $|\mathcal{V}| = 3T$, which indicates that $|\mathcal{V}| = 2T$ is likely a good choice that provides a good balance between prediction accuracy and computational complexity.

Finally, we consider all 1,000 test data points across the 10 different instances to compare the computation times of solving the original MILPs, the corresponding DFSOMs, and the original MILPs but only to the same objective function values as achieved by the DFSOMs. The third method serves as a benchmark heuristic solution method that allows a fair comparison with the DFSOM. Figure 4 shows the number of instances solved by each of the three methods as the solution time increases. One can see that the surrogate LP clearly outperforms the MILP, even when the MILP is only solved to achieve a solution of the same quality as the DFSOM.

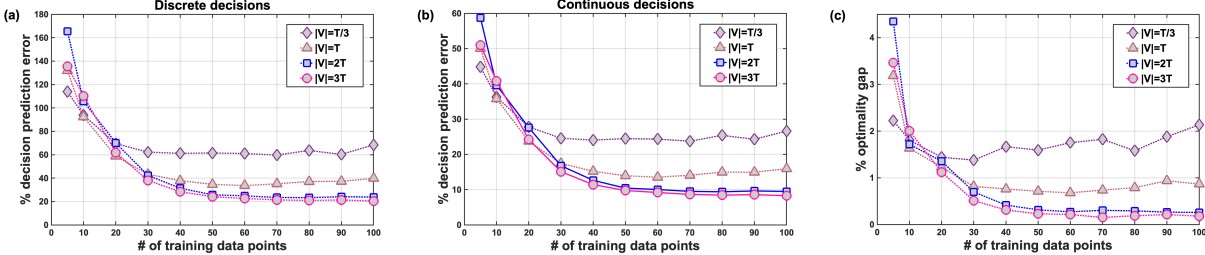

Figure 3: DFSOM performance for varying number of constraints learned ($|\mathcal{V}|$) in the hybrid vehicle control problem.

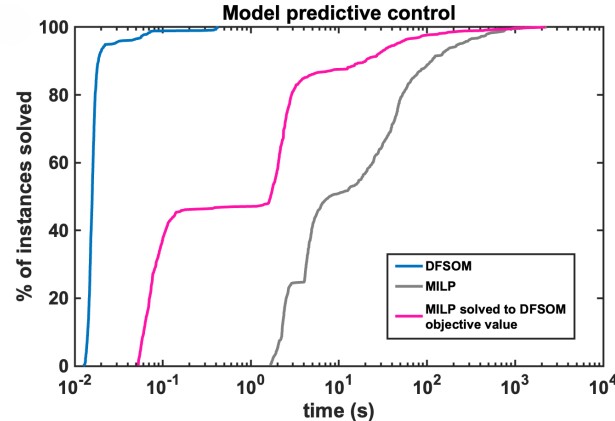

Figure 4: Computational performance of DFSOMs evaluated across 1,000 random instances in the hybrid vehicle control case compared to directly solving the original MILPs and solving the MILPs to the same objective values achieved by the corresponding DFSOMs

.

## 4.2 Production scheduling

In the second case study, we consider a single-stage production scheduling problem with parallel units, which can be formulated as the following MILP (Maravelias, 2021):

$$\underset{S,\,T,\,X}{\text{minimize}} \quad \sum_{i \in \mathcal{I}} \sum_{j \in \mathcal{J}} \sum_{t \in \mathcal{T}} \gamma_{ij} \, X_{ijt} \tag{6a}$$

$$T_{jt} \geq T_{j,t-1} \quad \forall\, j \in \mathcal{J},\, t \in \mathcal{T} \tag{6b}$$

$$\sum_{j \in \mathcal{J}} \sum_{t \in \mathcal{T}} X_{ijt} = 1 \quad \forall\, i \in \mathcal{I} \tag{6c}$$

$$\sum_{i \in \mathcal{I}} X_{ijt} \leq 1 \quad \forall\, j \in \mathcal{J},\, t \in \mathcal{T} \tag{6d}$$

$$S_{ij} \leq M \sum_{t \in \mathcal{T}} X_{ijt} \quad \forall\, i \in \mathcal{I},\, j \in \mathcal{J} \tag{6e}$$

$$S_{ij} \geq T_{j,t-1} - \eta(1 - X_{ijt}) \quad \forall\, i \in \mathcal{I},\, j \in \mathcal{J},\, t \in \mathcal{T} \tag{6f}$$

$$S_{ij} + \tau_{ij} \leq T_{jt} + \eta(1 - X_{ijt}) \quad \forall\, i \in \mathcal{I},\, j \in \mathcal{J},\, t \in \mathcal{T} \tag{6g}$$

$$\sum_{j} S_{ij} \geq \rho_i \quad \forall\, i \in \mathcal{I} \tag{6h}$$

$$\sum_j S_{ij} + \sum_{j \in \mathcal{J}} \sum_{t \in \mathcal{T}} \tau_{ij} X_{ijt} \le \epsilon_i, \quad \forall i \in \mathcal{I} \tag{6i}$$

$$S_{ij} \ge 0, \ T_{jt} \ge 0, \ X_{ijt} \in \{0,1\} \quad \forall i \in \mathcal{I}, j \in \mathcal{J}, t \in \mathcal{T}, \tag{6j}$$

where $\mathcal{J}$ denotes the set of units, $\mathcal{I}$ is the set of batches to be processed, $\mathcal{T}$ is the set of time slots, and $\eta$ is the horizon length. For each batch $i$, we are given a release time $\rho_i$, a due time $\epsilon_i$, and processing time $\tau_{ij}$ if it is processed on unit $j$. The binary variable $X_{ijt}$ equals 1 if batch $i$ is processed on unit $j$ in time slot $t$, which is associated with a cost denoted by $\gamma_{ij}$; $S_{ij}$ denotes the start time of batch $i$ on unit $j$, and $T_{jt}$ is the end time of time slot $t$ for unit $j$. The objective is to minimize the total production cost given in (6a). Constraints (6b) ensure the non-negative length of each time slot, equations (6c) ensure that every batch is processed, (6d) allow at most one batch to be processed on a given unit at a given time, (6e) set the start time of a batch on a unit to 0 if the batch is not being processed at any time on that unit and otherwise provide a bound on the start time, (6f) and (6g) combined determine the end time of a time slot on each unit, (6h) only allow a batch to start processing after its release, and (6i) ensure that the processing of each batch is completed before its due time.

### 4.2.1 Data generation and surrogate design

In all the 10 randomly generated problem instances, we use a horizon length of $\eta = 40$. We choose $\tau_{ij} \sim \mathcal{N}(1,7)$, $\rho_i \sim \mathcal{N}(1,9)$, $\epsilon_i \sim \mathcal{N}(10,39)$, $\gamma_{ij} \sim \mathcal{N}(30,100)$, and we set $M = \max_{i \in \mathcal{I}}(\epsilon_i)$. Here, the model input parameters are $\rho$, $\epsilon$, and $\tau$, which we vary to generate the training datasets. We also set $\mathcal{T} = \{1, \ldots, 8\}$, which is the smallest set that is still sufficiently large for all considered instances. With $|\mathcal{I}| = 13$, $|\mathcal{J}| = 4$, and $|\mathcal{T}| = 8$, the MILP has 84 continuous variables, 416 binary variables, and 987 constraints.

In the hybrid vehicle control case study, we learned additional linear inequalities that involve all decision variables. If we were to take the same approach here, each new constraint would involve $|\mathcal{I}||\mathcal{J}||\mathcal{T}| + |\mathcal{I}||\mathcal{J}| + |\mathcal{J}||\mathcal{T}|$ decision variables; this would lead to a very large number of parameters to be learned, which would significantly increase the computational complexity of the DFSM problem. Hence, instead, we try to exploit the structure of the scheduling problem in designing the additional constraints. Specifically, we notice that most of the constraints in (6) are written for all $i \in \mathcal{I}$, and it also makes intuitive sense that the relationships are strongest among the variables associated with the same batch. Using this rationale, we propose to add for each batch $i$ linear inequalities that only involve variables associated with that batch. The constraint coefficients and right-hand-sides are given as quadratic functions of the input parameters. In this case, the time taken to train the surrogate model for 100 training data points is around 3,000 seconds.

### 4.2.2 Surrogate model performance

Like in the first case study, we assess the performance of the DFSOM and the three NN-based optimization proxies. Note that the MILP solutions that serve as the basis for comparison are obtained at 1% optimality gap since many problem instances cannot be solved to full optimality within several hours. The results in Figure 5 show that the DFSOM clearly outperforms the NN, NN+LD, and GNN. Here, unlike in the first case study, the GNN's performance does not come close to that of the DFSOM, which could be attributed to the significant increase in problem size with this production scheduling problem having 500 decision variables and close to 1000 constraints. Moreover, the training time for GNN here is an order of magnitude higher compared to the DFSM approach as there are many more parameters to be learned in the GNN. This further showcases the advantage of DFSM where it is straightforward to leverage domain knowledge to reduce the size of the learning problem by choosing which variables to include in the additional constraints.

The DFSM approach is again very data-efficient, achieving a decision prediction error with respect to binary variables of less than 10% with only 50 training data points (Figure 5(a)). In this case, the prediction error with respect to the continuous variables is much higher and does not improve with increasing number of training data points (Figure 5(b)), yet the DFSOM still achieves very small optimality gaps (Figure 5(c)). The reason is that the objective function in problem (6) only involves the binary variables, which is why the optimality gap will be small as long as the solution in terms of the binary variables is predicted accurately. In addition, for a fixed assignment of batches to units and time slots (discrete decisions), there are many start times (continuous variables) that are feasible and hence optimal since they do not affect the cost. Due

to this multiplicity of optimal solutions, it is not surprising that the surrogate LP does not achieve the same solutions in terms of the continuous variables. But as long as the predicted solutions in terms of the binary variables are close to the true ones, the DFSOM will perform well. The difference between the surrogate solutions and the solutions obtained post feasibility restoration here is 8.3% when DFSOM is used, while 50.4% and 56.6% are achieved by the standard and Lagrangian-based NNs, respectively. The GNN shows some improvement with the difference reaching 38.1%.

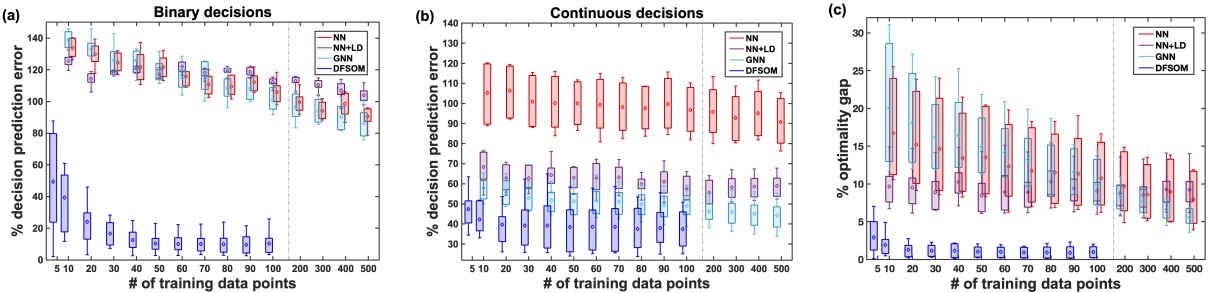

Figure 5: DFSOM performance compared to NN-based optimization proxies in the production scheduling problem.

Figure 6 shows the impact of the number of added constraints on the DFSOM's performance. Here, $|\mathcal{V}|$ denotes the number of constraints added per batch. Interestingly, we observe that it is sufficient to add just one constraint for each batch. In fact, the DFSOMs for $|\mathcal{V}| = 5$ and $|\mathcal{V}| = 10$ have much larger prediction errors with respect to the binary variables and optimality gaps for smaller training datasets, which indicates overfitting.

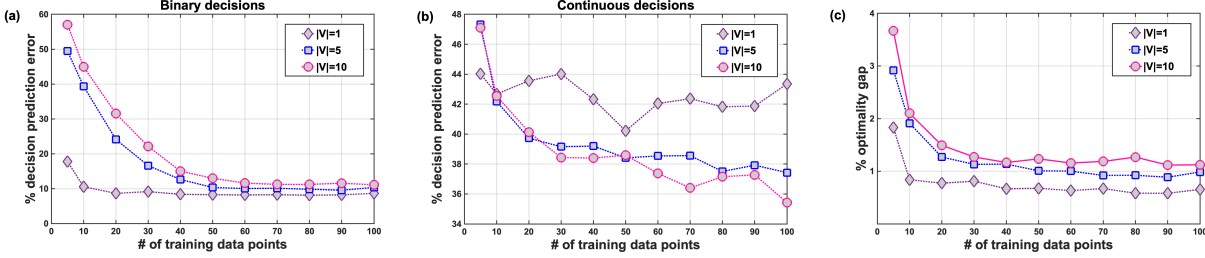

Figure 6: DFSOM performance for varying number of constraints learned ($|\mathcal{V}|$) in the production scheduling problem.

We again solve the 1,000 problem instances taken from the 10 different instances to compare the computation times of solving the original MILPs, the corresponding DFSOMs, and the original MILPs but only to the same objective function values as achieved by the DFSOMs. In this case, the original MILPs were solved to 1% optimality gap as it would have taken excessively long to solve them to full optimality. Even with 1% optimality, as shown in Figure 7, some instances of the MILP take multiple hours to solve while the DFSOM solves within a few seconds in all instances.

In addition, we test the performance of the DFSOM on some hard problem instances for which the corresponding MILPs could not be solved to 1% optimality within the given time limit of 3,000 seconds. We use 30 such hard instances to form the test dataset, while the DFSOM is trained only using instances that could be solved to optimality. The results are shown in the performance plot in Figure 8, where we observe that the DFSOM obtains integer-feasible solutions significantly faster when compared to solving the MILP to the same objective function values. We should note that our hope here was that the DFSOM might occasionally achieve even better solutions than the MILP (at the time limit) since it was trained using optimal solutions. While this does not seem to be true in this case, it is an aspect we plan to explore more in our future work.

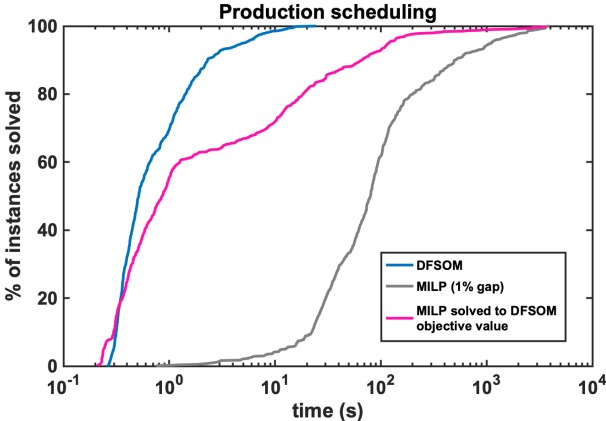

Figure 7: Computational performance of DFSOMs evaluated across 1,000 random instances in the production scheduling case compared to directly solving the original MILPs and solving the MILPs to the same objective function values achieved by the corresponding DFSOMs.

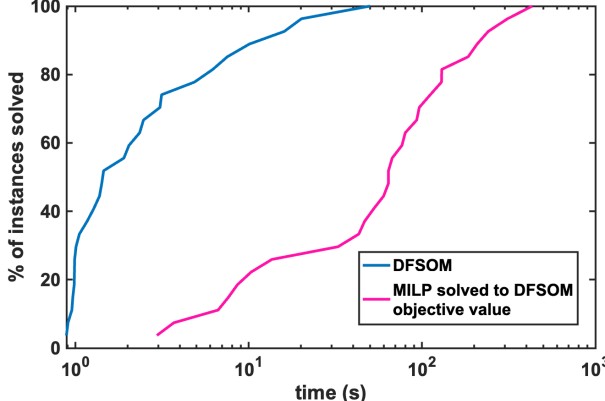

Figure 8: Computational performance of DFSOMs evaluated across 30 hard instances in the production scheduling case compared to solving the MILPs to the same objective function values achieved by the corresponding DFSOMs.

## 5    Conclusions

Motivated by the need to solve difficult MILPs in real-time settings, we developed a data-driven approach for constructing efficient surrogate LPs that can replace the MILPs. When generating these surrogate LPs, we explicitly try to minimize the decision prediction error defined as the difference between the optimal solutions of the surrogate and the original optimization problems. The resulting decision-focused surrogate modeling problem is a large-scale bilevel program that we solve using a penalty-based block coordinate descent algorithm. In our computational case studies, we demonstrated the efficacy of the proposed approach both in terms of prediction accuracy and data efficiency. It also allows the use of problem-specific knowledge to design surrogate model structures that enhance the training and/or the performance of the surrogate optimization models. The results further show that the surrogate LPs trained using the proposed approach outperform, often significantly, state-of-the-art NN-based optimization proxies.

## Acknowledgments

The authors gratefully acknowledge financial support from the National Science Foundation under Grant #2044077 and from the State of Minnesota through an appropriation from the Renewable Development Account to the University of Minnesota West Central Research and Outreach Center. They would also like to acknowledge the Minnesota Supercomputing Institute (MSI) at the University of Minnesota for providing resources that contributed to the research results reported in this paper.

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

# A   Appendix

## A.1   Illustrative example

To illustrate the working of the surrogate LP model, we present a 2-dimensional multi-constraint knapsack problem with one of its constraints changing as a function of the input parameter $u$:

$$\begin{aligned}
\underset{x_1 \in \mathbb{Z},\, x_2 \in \mathbb{Z}}{\text{maximize}} \quad & 4.8x_1 + 6x_2 \\
& 4x_1 + 3x_2 \le 70 \\
& 100ux_1 + 85x_2 \le 800u + 680 \\
& x_1 \le 17,\ x_2 \le 17.
\end{aligned} \tag{7}$$

We obtain an surrogate LP for problem (7) by adding linear inequalities to the original problem whose parameters are trained by solving the DFSM problem (3). The resulting DFSOM is the following LP:

$$\begin{aligned}
\underset{x_1 \in \mathbb{R},\, x_2 \in \mathbb{R}}{\text{maximize}} \quad & 4.8x_1 + 6x_2 \\[4pt]
& 4x_1 + 3x_2 \leq 70 \\
& 100ux_1 + 85x_2 \leq 800u + 680 \\
& (17.9 - 56.9u + 45.1u^2)x_1 + (-4.5 + 25.3u - 14.3u^2)x_2 \leq 100 \\
& (7.4 + 6.4u - 7.9u^2)x_1 \leq 100 \\
& (-0.2 + 14.5u - 9.4u^2)x_1 + (12 - 9.7u + 3.7u^2)x_2 \leq 100 \\
& x_1 \leq 17, \; x_2 \leq 17.
\end{aligned} \tag{8}$$

In Figure 9, we show for different values of $u$ the feasible regions of the LP relaxation of problem (7), which we denote as $\bar{\mathcal{S}}$, in blue. The feasible region of problem (8), denoted by $\mathcal{S}'$, is shown in green. The integer-feasible points for problem (7) constitute the set $\mathcal{S}$. From the figure, one can see that the solutions obtained by solving the surrogate LP (8) for different values of $u$ are the same as those obtained by solving the original MILP (7).

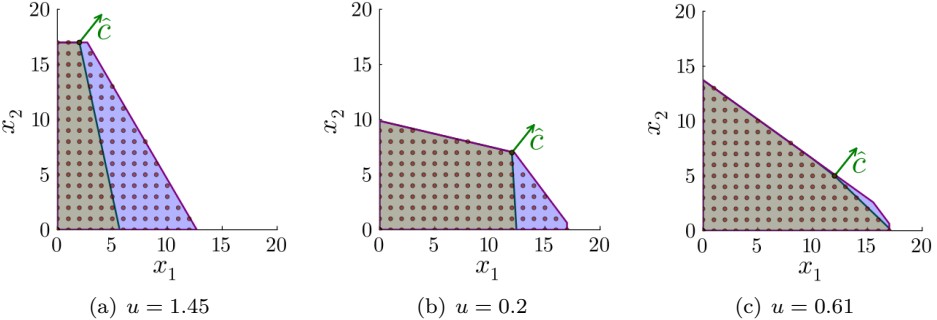

(a) $u = 1.45$      (b) $u = 0.2$      (c) $u = 0.61$

Figure 9: Comparison of feasible regions and optimal solutions of (7) and (8). For the three instances of $u = \{1.45; 0.2; 0.61\}$, the corresponding optimal solutions are $x^* = \{(2, 17); (12, 7); (12, 5)\}$.

Note that the added inequalities $\tilde{A}(u)x \leq \tilde{b}(u)$ may not be valid for all integer-feasible points of the original problem as DFSM only aims to minimize the prediction error with respect to the optimal solutions. Indeed, for the three cases of $u$-values shown in Figure 9, $\text{conv}(\mathcal{S}) \not\subseteq \mathcal{S}'$ in the first two instances as shown in Figures 9(a) and 9(b).

## A.2 The BCD algorithm

To describe the BCD approach, we first re-write the formulation (4) in the following form:

$$\begin{aligned}
\underset{\theta \in \Theta,\, \tilde{x},\, \lambda,\, \mu}{\text{minimize}} \quad & \sum_{i \in \mathcal{I}} \|x_i^* - \tilde{x}_i\|_1 \\[4pt]
\text{subject to} \quad & c + {A_i'}^T d_i' = 0 \quad \forall i \in \mathcal{I} \\
& A_i' \tilde{x}_i - b_i' \leq 0 \quad \forall i \in \mathcal{I} \\
& D(d_i')(A_i' \tilde{x}_i - b_i') = 0 \quad \forall i \in \mathcal{I} \\
& d_i' \geq 0, \; \tilde{x}_i \in \mathbb{R}^n \quad \forall i \in \mathcal{I} \\
& A_i' = \begin{bmatrix} A(\bar{u}_i) \\ \tilde{A}(\bar{u}_i, \theta) \end{bmatrix}, \; b_i' = \begin{bmatrix} b(\bar{u}_i) \\ \tilde{b}(\bar{u}_i, \theta) \end{bmatrix}, \; d_i' = \begin{bmatrix} \lambda_i \\ \mu_i \end{bmatrix} \quad \forall i \in \mathcal{I}.
\end{aligned} \tag{9}$$

Problem (9) is a nonconvex NLP which becomes intractable for large instances. Furthermore, some constraints, such as the complementarity constraints, in the problem cause it to violate constraint qualification conditions leading to convergence difficulties for standard NLP solvers.

We address the lack of regularity in (9) by considering its penalty reformulation in (10), which is a standard strategy employed in the NLP literature for degenerate NLPs such as (9):

$$
\underset{\theta \in \Theta, \, \tilde{x}, \, d'}{\text{minimize}} \quad \sum_{i \in \mathcal{I}} \|x_i^* - \tilde{x}_i\|_1 + q^T \begin{bmatrix} \sum_{i \in \mathcal{I}} \|c + {A_i'}^T d_i'\|_1 \\ \sum_{i \in \mathcal{I}} \max\{0, A_i' \tilde{x}_i - b_i'\} \\ \sum_{i \in \mathcal{I}} \|d_i'^T (A_i' \tilde{x}_i - b_i')\|_1 \end{bmatrix} \tag{10}
$$
$$
\text{subject to} \quad d_i' \geq 0, \, \tilde{x}_i \in \mathbb{R}^n \quad \forall \, i \in \mathcal{I},
$$

where we penalize the violation of the constraints using a nonsmooth $\ell_1$-norm-based penalty function. This particular penalty function is known to be exact in the sense that, for values of the penalty parameters $q$ larger than a certain threshold, every optimal solution of (9) also minimizes (10). Therefore, instead of (9), one can solve (10) which is amenable to solution via standard NLP solvers if the description of the set $\Theta$ satisfies necessary constraint qualification conditions.

However, the threshold value for $q$ above which the exactness of the penalty reformulation holds is generally hard to determine a priori. Therefore, we employ an iterative approach where we start with small values for $q$ and successively increase them by a factor $\rho$ in every iteration until we arrive at a solution that is feasible for (9) (Algorithm 1).

The reformulated problem (10) is still a nonconvex NLP that is difficult to solve when the number of data points ($|\mathcal{I}|$) is large.. We alleviate this issue by observing that (10) is a multiconvex optimization problem if the set $\Theta$ is convex. This feature of (10) allows us to solve it via an efficient block-coordinate-descent (BCD) approach. While implementing BCD, we tackle the nonsmoothness of the objective functions of BCD subproblems by augmenting them with the proximal operator.

---

**Algorithm 1** The penalty-based BCD algorithm.

---

initialize $(\tilde{x}, \theta, d')$ with a feasible solution from IPOPT
initialize $q \leftarrow q^0$, fix $\rho \leftarrow \bar{\rho}$
**while** convergence criteria for blocks are not satisfied   **do**
  solve (10) with BCD or IPOPT,
$$
q = q + \bar{\rho} \begin{bmatrix} \sum_{i \in \mathcal{I}} \|c + {A_i'}^T d_i'\|_1 \\ \sum_{i \in \mathcal{I}} \max\{0, A_i' \tilde{x}_i - b_i'\} \\ \sum_{i \in \mathcal{I}} \|d_i'^T (A_i' \tilde{x}_i - b_i')\|_1 \end{bmatrix}
$$
**end while**

---

## A.3   Design of NN-based optimization proxies

In our case studies, we contrast the performance of the proposed DFSOM approach with three different NN-based optimization proxies as surrogates for the MILPs. In the following, we briefly outline the design and training of these NNs. Note that there are no hard constraints embedded in these NN models; hence, the feasibility of the predicted optimal solutions with respect to the constraints of the original model is not guaranteed. Therefore, we use the same feasibility restoration strategy that is applied to the outputs of the DFSOM to obtain integer-feasible solutions from the NN-based optimization proxies; a discussion of this strategy is provided in the next section.

### A.3.1   Feedforward NN architecture

To train the NN models, we split the data containing input parameters and optimal solutions into training and test datasets. We use the mean squared error between the true and predicted values as the loss function and apply the ADAM optimizer to estimate the NN parameters. Similar to DFSOM, for a given input, the NN model is also trained to predict an optimal solution for the original problem.

For the feedforward NN, the numbers of hidden layers and neurons in each layer are determined through a trial-and-error approach aiming to minimize the training loss over a fixed number of epochs. The number of hidden layers is based on the empirical observation that increasing the number of hidden layers resulted in little improvements in the model's accuracy. The design of the specific NN model used in the hybrid vehicle control case study is as follows: the input layer contains $T$ nodes, the same as the dimension of the input vector $u$. The two hidden layers have $3T$ and $6T$ nodes, and finally, the output layer consists of $3T + 1$ nodes aligning with the dimensionality of the decision space. We also account for the effect of different learning rates (namely 0.1, 0.01, and 0.001) on the training loss. Based on our analysis, we determine 0.01 to be the most suitable learning rate for our purpose.

### A.3.2 Augmented Lagrangian-based NN architecture

Here, the NN has the same feedforward architecture as described above, but the loss function is augmented with penalty terms associated with the Lagrangian dual of the optimization problem. The Lagrange multipliers are updated using a subgradient approach in each iteration of the NN training process as discussed in Van Hentenryck (2025). Assume the given MILP is of the following form:

$$
\begin{aligned}
\underset{x}{\text{minimize}} \quad & c(u)^\top x \\
& a_i(u)^\top x \le b_i(u) \quad \forall i \in \mathcal{I} \\
& g_e(u)^\top x = h_e(u) \quad \forall e \in \mathcal{E} \\
& x \in \mathbb{R}^m \times \mathbb{Z}^{n-m}.
\end{aligned}
\tag{11}
$$

Then, the augmented loss function can be written as follows:

$$
\mathcal{L}^d_{\lambda,\mu}(y,\hat{y}) = \|y - \hat{y}\| + \sum_{e \in \mathcal{E}} \lambda_e |g_e(u)^\top \hat{y} - h_e(u)| + \sum_{i \in \mathcal{I}} \mu_i, \max\{0, a_i(u)^\top \hat{y} - b_i(u)\},
\tag{12}
$$

where $\hat{y}$ are the NN model predictions given by $\hat{y} = \mathcal{M}_\theta(u)$, and the model parameters $\theta$ are estimated in each epoch by keeping the Lagrange multipliers fixed from the last iteration.

After these model parameters are tuned from $\theta^t \to \theta^{t+1}$, the multipliers are adjusted using the constraint violations as follows:

$$
\begin{aligned}
\lambda_e^{t+1} &= \lambda_e^t + \rho \frac{1}{n} \sum_{k=1}^{n} |g_e(u_k)^\top \mathcal{M}_{\theta^{t+1}}(u_k) - h_e(u_k)|, \\
\mu_i^{t+1} &= \mu_i^t + \rho \frac{1}{n} \sum_{k=1}^{n} \max(0, a_i(u_k)^\top \mathcal{M}_{\theta^{t+1}}(u_k) - b_i(u_k)).
\end{aligned}
\tag{13}
$$

### A.3.3 GNN architecture

Here, we leverage the strong representation power of GNNs to capture the structure of MILPs. We use a weighted bipartite graph consisting of variables nodes, constraints nodes, and edges, where an edge connects a variable node with a constraint node if the variable appears in that constraint. For example, problem (7) of the illustrative example can be represented as the graph shown in Figure 10.

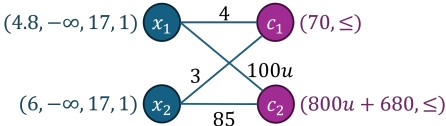

Figure 10: Bipartite graph representation of the MILP (7) with variable nodes on the left and constraint nodes on the right.

The GNN incorporates three sets of features:

1. The features of each variable node contain information about the correspponding coefficient in the objective function, the lower and upper bounds of that variable, and whether the variable is discrete (1) or continuous (0).

2. The features of each constraint node contain information about the corresponding right-hand-side value and whether the constraint is an inequality ($\leq$) or an equality ($=$).

3. The edge weights are the coefficients of the variables in the constraints that they appear in.

Using this graph representation of the MILP as input, we train a graph convolutional neural network as described in Gasse et al. (2019) to predict the optimal solutions.

## A.4  Feasibility restoration

The predicted decisions from the DFSOM may not always satisfy the integrality constraints. For such instances, we employ an inexpensive feasibility restoration strategy to obtain an integer-feasible solution from the predicted solution. Let $\tilde{x}_i$ be the optimal solution of the DFSOM for $\bar{u}_i$. Suppose the discrete decisions associated with $\tilde{x}_i$, denoted by $\tilde{x}_i^z$, are not integer. To restore integer feasibility, we solve the following problem:

$$\underset{x \in \mathbb{R}^m \times \mathbb{Z}^{n-m}}{\text{minimize}} \quad \|x^z - \tilde{x}_i^z\|_1$$
$$A(\bar{u}_i)x \leq b(\bar{u}_i). \tag{14}$$

We then solve the original MILP with the discrete decision variables fixed to the values of the optimal solution to problem (14) to obtain the corresponding optimal values for the continuous variables. Figure 11(a) provides an illustration of an infeasible prediction from the DFSOM. By using the feasibility restoration strategy described above, we obtain the closest integer-feasible point as shown in Figure 11(b) .

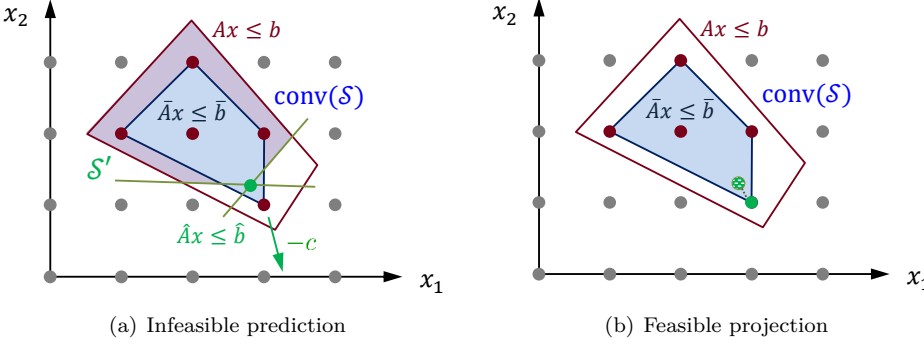

(a) Infeasible prediction $\qquad\qquad$ (b) Feasible projection

Figure 11: The predicted decision variables that are not integer-feasible are used in a projection problem to return discrete decision values.

## A.5  Additional results

For the hybrid vehicle control problem, Figures 12 and 13 compare the performance of the DFSOM and the NN-based optimization proxy for $T = 10$ and $T = 20$, respectively. The DFSOMs were trained with $|\mathcal{V}| = 2T$. The results indicate DFSOM's superior performance even for these smaller problems, emphasizing the benefit of incorporating original constraints in DFSM. Notably, with only 50 training data points, DFSOM reliably yields solutions with an optimality gap close to zero in both cases.

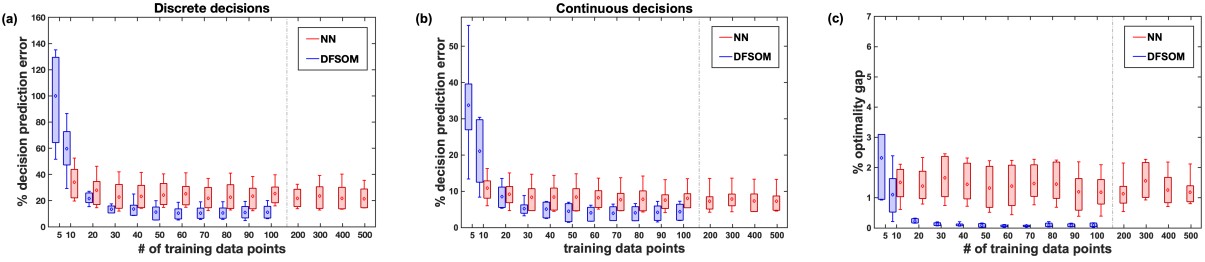

Figure 12: DFSOM performance compared to NN-based optimization proxy for $T = 10$.

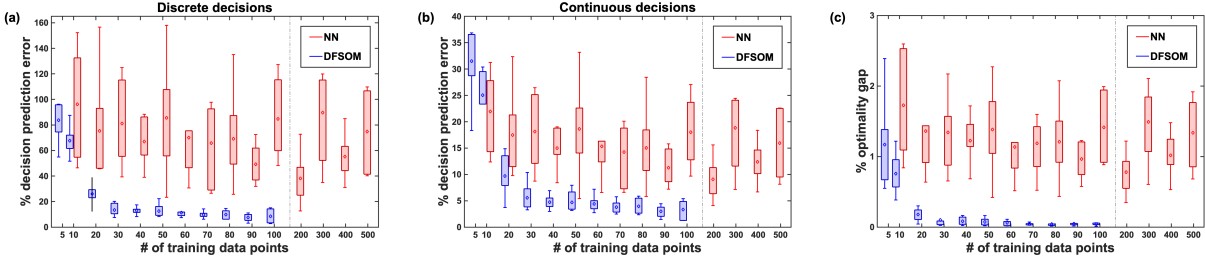

Figure 13: DFSOM performance compared to NN-based optimization proxy for $T = 20$.

