# OpenReview forum: "Decision-Focused Surrogate Modeling for Mixed-Integer Linear Optimization"
_TMLR — Accepted by TMLR_

### Review · Reviewer_NpQz · 2024-12-27

**Summary Of Contributions:**

Consider a mixed-integer linear program (MILP) in the form:
$$\begin{split} \operatorname*{minimize}_x c(u)^{\top}x \\\\ A(u)x \leq b(u) \\\\ x \in \mathbb{R}^m\times \mathbb{Z}^{n-m} \end{split} \tag{1}$$
where $u$ is an input parameter. Suppose that this MILP needs to be solved repeatedly, for different values of $u$. As is well-known, solving an MILP can be computationally expensive, whereas solving a Linear Program (LP) is fairly cheap. Thus, this paper considers the problem of _learning_ LP surrogates, parametrized by $u$, for a parametrized MILP as above.

The surrogate LPs are constructed by dropping the integrality constraints in (1) and adding additional, learned linear inequality constraints: $\hat{A}(u,\theta)x \leq \hat{b}(u,\theta)$. Here $\theta$ represents parameters encoding the functional form of the learned constraints.  The parameters $\theta$ are learned in a data-driven way, from a training set of parameter-solution pairs, $(u_i,x_i^*)$, where $x_i^*$ represents the solution to (1) given $u = \bar{u}_i$.  An innovative optimization scheme is proposed to do so.

The paper is rounded out by two interesting case studies, both of which involve challenging MILPs. The results of both case studies indicate that the proposed approach works well---providing fast solutions with small optimality gap.

**Audience:**

Yes

**Claims And Evidence:**

Yes

**Requested Changes:**

See above.

**Strengths And Weaknesses:**

### Strengths
 - The paper is well-written and clear
 - The idea of "learning cuts" for turning MILPs into LPs is intuitive, yet novel in this context.
 - The experiments are good. I appreciate the use of problems grounded in real world scenarios.

### Weaknesses
 - I think the learning problem/solution strategy this paper hones in on (i.e., $\ell_1$ loss function, use KKT conditions to turn a bilevel problem into a single level, constrained problem) is a bit narrow. I think this paper could benefit from further discussion of alternate approaches (e.g., using the $\ell_2$ loss: $\\|x_i^* - \hat{x}_i\\|_2^2$).

### Minor points/questions
 1. Throughout the paper, the abbreviations DFSM and DFSOM are used, but only DFSM is defined. Is this a typo, do both abbreviations reference the same thing, or is there a subtle difference?
 2. As mentioned on pg. 5, you use low-order polynomials to parametrize the added constraints, and I agree that they work well in your case studies. Did you try any other parameterization schemes? Would you suggest other parameterization schemes to practitioners interested in implementing your framework?
 3. (minor) Equation 2 is split over two pages. If you can fiddle with the formatting to get it onto one page, that might be nice.
 4. The use of hats to denote LP solutions ($\hat{x}$) as well as added constraints  ($\hat{A},\hat{b}$) is confusing. I suggest using $\tilde{A},\tilde{b}$ instead.
 5. Why do you use $\ell_1$ error in equation (3), as opposed to, e.g., $\ell_2$?
 6.  For both DFSM and the NN approach, a feasibility restoration step is applied (as discussed in Section 4.1.2). Could the authors comment on how close to integral solutions from both methods tend to be?

---

> ### Author Response · Authors · 2025-01-25
> **Response to Reviewer NpQz's comments**
>
> We thank the reviewer for the valuable feedback! Responding to the reviewer’s comment under “weaknesses,” we agree that the learning problem could have been solved using alternative methods and we could also have considered different variants of the problem itself (e.g. by changing the loss function). Indeed, we did explore some of these alternatives in the initial phase of the project but eventually converged to the presented solution method, which worked well in our computational experiments. Since the focus of this work was to introduce a new surrogate modeling approach and demonstrate the potential benefits of that approach, we found it sufficient to have an algorithm that could solve the problem and hence did not further provide a systematic comparison of different solution methods. However, this certainly will be part of our future work as we’re attempting higher-dimensional problems for which more efficient solution algorithms may need to be developed. Regarding the loss function, we did try both the $\ell_1$ and $\ell_2$ norms in our initial experiments but did not observe a notable difference. This is further indicated by the computational results where in most instances, a zero loss was achieved, which means that the predicted decisions take the same value as the observed decisions. In that case, either loss function would have achieved the same solution (assuming the solution is unique); hence, upon that observation, we did not further investigate other loss functions.
>
> Our responses to the minor points/questions raised by the reviewer are listed below:
>
> 1. We thank the reviewer for bringing this to our attention. To clarify, DFSM stands for the “decision-focused surrogate modeling” approach, while DFSOM refers to the resulting “decision-focused surrogate optimization model.” These abbreviations represent closely related but distinct things. We will expand DFSOM at its first mention and revise the manuscript to ensure this distinction is clearly communicated.
>
> 2. NNs could be a good alternative for predicting the constraint coefficients in more complex problems given their generalizability and strong representation power. However, incorporating NNs would make the learning problem much more difficult to solve as NNs would likely involve a significantly larger number of trainable parameters and possibly require a mixed-integer representation (e.g. if ReLU activation functions are used). We plan to investigate this in our future work.
>
> 3. Thanks for pointing this out. We will address this formatting issue for equation 2.
>
> 4. We appreciate the reviewer’s suggestion and will revise the notation to improve clarity. Specifically, we will update the use of hats for LP solutions ($\hat{x}$) and replace the hats on the constraints ($\hat{A}, \hat{b}$) with bars ($\bar{A}, \bar{b}$) to distinguish between the two more clearly.
>
> 5. During our initial experiments, we tested both $\ell_1$ and $\ell_2$ norms and did not observe a notable difference in performance. The computational results showed that, in most cases, the predicted decisions matched the observed decisions when the $\ell_1$ norm was used, suggesting that both loss functions would have yielded similar outcomes. Based on these findings, we opted for the $\ell_1$ norm, but our method is not restricted to this choice.
>
> 6. This is a great point! We will add that information to the manuscript. Specifically, for the hybrid vehicle control case study, the difference between the surrogate solutions and the solutions obtained post feasibility restoration is 6.6% with a standard deviation of 2.4% when DFSOM is used as a surrogate. That difference increases to 26.82% with a standard deviation of 10.8%  when NN based optimization proxy is used as a surrogate. A total of 1000 instances are considered to evaluate this difference. For each of these instances, the decision vector is scaled in continuous decisions to match the range of discrete decisions in order to generate fair comparisons between the predicted surrogate decisions and those obtained after feasibility projection. Moreover, the majority of the % difference mentioned stems from the discrete decisions changing after the feasibility restoration step.

---

> > ### Comment · Reviewer_NpQz · 2025-01-27
> > **Response to Response to Reviewers Comments**
> >
> > Thanks to the authors for their thorough response. I'd recommend adding a sentence or two to the paper on the choice of loss function.

---

> > > ### Author Response · Authors · 2025-01-27
> > > **Response to Reviewer NpQz's response**
> > >
> > > Thank you for the quick response and suggestion! We will certainly do that, i.e. add a remark on the choice of loss function.

---

### Review · Reviewer_ujd8 · 2024-12-30

**Summary Of Contributions:**

The authors propose a novel decision-focused modeling strategy to approximate the solution of MILPs, offering a method that is interpretable, data-efficient, and computationally inexpensive during inference. By leveraging surrogate linear programs (LPs) to approximate the structure of MILPs, the approach achieves near-optimal solutions while significantly reducing computation time.

**Audience:**

Yes

**Claims And Evidence:**

No

**Requested Changes:**

Some benchmarks as mentioned in the weaknesses section.

**Clarifications:**
- In Section 4.1.1, Does the training dataset include several D for a single draw of parameters? Will this violate the i.i.d assumption?
- DFSOM & NLP are not expanded

**Strengths And Weaknesses:**

**Strengths**
1. The proposed method is data-efficient, requiring relatively fewer instances compared to neural network proxies. However, further exploration of additional capabilities achievable with the method could enhance its impact (see weaknesses).

2. The proposed method shows potential for application in online settings, which is a common requirement in real-world scenarios. However, this potential should be further clarified (see weaknesses).

3. The method effectively solves large instances of DFSM by leveraging the problem's multiconvexity and employing block coordinate descent (BCD), ensuring computational tractability for complex problems.

**Weaknesses**

1. While the authors suggest that the method is applicable to online settings, it is unclear what this entails. From the experiments, some problems are solved in approximately 100 seconds, which, depending on the domain, might align more with offline settings. To manage expectations about where the method can be effectively used, it would be beneficial to provide a clearer definition of "online setting" (e.g., in the introduction) and its relevance to the application context. It is worth acknowledging that solving within 100 seconds could still be considered "online" in certain domains. Highlighting such domains explicitly would strengthen the claim. However, a key concern is whether the cost of solving an exact MILP or another expensive optimization approach in these domains makes the proposed method sufficiently competitive. Exploring a comparison of solving times and costs across these settings would provide a more comprehensive evaluation of the method's practical utility.


2. A discussion of scenarios where the method might fail or perform suboptimally would strengthen the paper.
It would be interesting to see how the proposed method could benefit from using NNs to predict constraint cofficients instead of polynomial functions. For instance, if NNs were used to predict parameter coefficients, could the method learn better constraints and achieve improved generalizability?

3. The benchmarking results leave some room for improvement.
    - Graph Neural Networks (GNNs), which are well-suited for MILPs, should serve as a baseline for prediction tasks. While using MLPs is a reasonable choice, it is less common in the community for such tasks. Reassessing benchmarks using GNNs will provide a stronger evaluation and validation of the claims. However, it is important to note that GNNs are less interpretable and data-efficient than the proposed method, so the proposed method still holds relevance to the community.

    - While the related works section discusses ML-based optimization proxies, the paper does not directly compare its method with these approaches. Such comparisons would provide valuable insights into the method’s relative performance and unique advantages.


4. A standard practice is to test a method’s robustness and generalizability by evaluating performance on harder instances. This would help identify the boundaries of the method's applicability. For example, displaying geometric mean solving times across problem categories would offer a clearer picture of its performance across varying complexities.

---

> ### Author Response · Authors · 2025-01-25
> **Response to Reviewer ujd8's comments**
>
> We thank the reviewer for their constructive feedback. Please see below our responses to the reviewer’s comments listed under weaknesses.
>
> 1. For the first case study of hybrid vehicle control, all instances were solved under 1 s, which we believe would be a desired solution time for an application of this format. For the second case study of production scheduling, all instances were solved under 30 s, which is sufficient as the time between schedule updates is typically on the order of minutes or hours. In the scope of this work, the term “online setting” refers to applications where solutions are updated frequently by each time solving a new instance of the same optimization problem. We present case studies that can directly benefit from our proposed method and where the direct exact solution approach may lead to much greater solution times that delay the working of these online systems. We will provide more clarification on this aspect in the revised manuscript.
>
> 2. While we have considered cases where the proposed approach did not perform as well, we did not include a detailed analysis in the manuscript. Based on our observations, our approach may not be a good choice in the following situations: (1) The MILP is too easy to solve, offering no significant advantage for applying DFSM. (2) The MILP is too complex, making data collection for training impractical due to the need for optimal or near-optimal solutions. (3) Each learned inequality includes all variables which makes training the DFSOM difficult as the number of added cuts increases. We will add a remark on these limitations in the revised manuscript.
> Regarding the use of NNs to predict constraint coefficients, we completely agree with the reviewer that NNs, with their strong representational power, could offer better generalizability for complex problem instances. We simply did not see the need to apply them in our current case studies because polynomials turned out to perform very well already. We want to mention though that incorporating NNs would make the learning problem much more difficult to solve as NNs would likely involve a significantly larger number of trainable parameters and possibly require a mixed-integer representation (e.g. if ReLU activation functions are used). One could also apply frameworks like OptNet (Amos & Kolter, 2017) that can integrate optimization within NN architectures; however, they have their own scalability limitations. That being said, while we did not need to use NNs in this work, these are all directions that we are considering for our future work. We will add a remark addressing this in the revised manuscript.
>
> 3. (a) While GNNs are indeed powerful for prediction tasks in MILPs, they often require architecture designs that are specifically tailored to the problem structure, which can limit their generalizability across different problems. Furthermore, when dealing with problem instances that have similar structures but subtle, essential differences, GNNs may struggle to distinguish these differences accurately (Chen et al., 2022). To address this, it has been proposed to append random features at the node level to these graphs to break symmetry and enable GNNs to represent these instances separately, making the solution strategy highly instance-specific. This approach can undermine the goal of developing a generalizable and interpretable framework, which is a key advantage of our proposed method; hence, we did not opt to compare the DFSM method against GNNs.
> (b) In the current manuscript, we focused on NN-based optimization proxies as they are typically the first approach one would try before exploring more complex methods, making them a suitable baseline for comparison. Our primary objective was to introduce our new surrogate modeling approach and highlight its potential benefits. However, we agree that a broader comparison would strengthen the paper’s impact. To address this, we are willing to extend our computational results to include a state-of-the-art Lagrangean dual-based NN surrogate, which was initially proposed by Fioretto, Mak, and van Hentencryck in 2020. We would appreciate the reviewer’s guidance on whether this addition would be satisfactory or if they would suggest evaluating another method.
>
> 4. We appreciate the reviewer’s suggestion to evaluate the method’s robustness on harder instances. In response, we will include instances from the second case study that are significantly more challenging to solve. Initially, we excluded these instances from the paper because they could not be solved to optimality within a reasonable time frame, often only yielding suboptimal solutions. For a fair comparison, we refrained from reporting DFSM results for these instances as direct comparison with suboptimal solutions might be misleading. However, we will now include these instances, comparing the best solutions found within the time limit and the computation time required to reach them.

---

> > ### Comment · Reviewer_ujd8 · 2025-02-04
> > **thanks for the response**
> >
> > Thanks for clarifying these points. The discussion of points 1, 2, and 4 will strengthen the paper, and I appreciate the authors' solution approach. My only concern is regarding the benchmarking.
> >
> > Regarding point 3, I believe a GNN benchmark should be included for completeness. NNs do not encode the correct inductive biases for MILPs (e.g., permutation invariances among variables and constraints), so they should not be the default choice. Since the authors acknowledge the potential downsides of GNNs, it would be valuable to verify whether those concerns hold and, if so, whether appending the random vector mitigates them. Including such a benchmark does not undermine the proposed solution; rather, it provides a clearer comparison for the community.
> >
> > Additionally, a Lagrange dual-based NN surrogate would be a useful benchmark to consider.

---

> > > ### Author Response · Authors · 2025-02-11
> > > **Response to Reviewer ujd8**
> > >
> > > Thank you for the response! We've been working on the changes that we proposed in our previous response, and we should be able to get them done by the initial deadline. However, the inclusion of the GNNs would take some more time; we will ask the Action Editor if an extension could be granted for us to work on it.

---

> > > > ### Author Response · Authors · 2025-03-05
> > > > **Response to Reviewer ujd8**
> > > >
> > > > Following the reviewer's suggestion, we have now added results from GNN-based optimization proxies for both case studies. As the reviewer expected, it does indeed perform better than the other NN-based optimization proxies; however, it is still outperformed by the proposed DFSM approach, especially in the higher-dimensional case. We want to again thank the reviewer for their input! Implementing the GNNs has led to additional valuable insights and greater assurance that our proposed approach is competitive to the state of the art.

---

> > > > > ### Comment · Reviewer_ujd8 · 2025-03-05
> > > > > **Thank you!**
> > > > >
> > > > > Thanks for reporting your results on GNNs. As expected, it improves the performance, but it is still not data efficient. I am satisfied about the merits of the proposed approach.

---

### Review · Reviewer_RhjW · 2025-01-15

**Summary Of Contributions:**

This paper proposes a data-driven approach for online mixed-integer linear programs (MILPs), which learns surrogate linear programs (LPs) by adding cutting planes, i.e., linear inequalities, that can be solved more efficiently and yield similar optimal solutions compared to the original MILPs. Instead of learning new cutting planes whenever new parameters of the MILP are provided, the approach learns the parameters of the cutting planes as functions of the MILP parameters. Thus, when given a set of new MILP parameters, the learned functions automatically provide a new set of parameters for the cutting planes. The authors use the l1-norm to measure the difference between the LP solution and the MILP solution, first formulating the learning task as a bilevel optimization problem and then reformulating it into a single-level optimization problem by replacing the lower-level problems with their KKT optimality conditions. Computational case studies on hybrid vehicle control and production scheduling demonstrate that DFSOM outperforms NN-based optimization proxies in terms of prediction accuracy, data efficiency, and computational performance.

**Audience:**

Yes

**Claims And Evidence:**

Yes

**Requested Changes:**

Please refer to "Weaknesses".

**Strengths And Weaknesses:**

Strengths:
1. The idea of interpreting the learning of LP surrogates as learning parametric cutting planes is interesting and novel.
2. The paper is generally well-written and clear.

Weaknesses:
1. While the paper compares DFSOM to NN-based optimization proxies, it does not evaluate their performance against SOTA, i.e., other ML-based optimization methods, limiting the broader impact of the findings and their positioning within the existing body of literature.
2. Conducting experiments on realistic datasets would be beneficial.
3. While the penalty-based BCD algorithm provides an efficient solution for training the surrogate models, its reliance on iterative optimization could become computationally expensive for very large MILPs. Additionally, the feasibility restoration process, used to correct infeasible predictions, adds computational overhead that may not be trivial in time-sensitive applications.

---

> ### Author Response · Authors · 2025-01-25
> **Response to Reviewer RhjW's comments**
>
> We thank the reviewer for their constructive feedback. Please see below our responses to the reviewer’s comments listed under weaknesses.
>
> 1. We appreciate the reviewer’s suggestion to compare our approach with state-of-the-art (SOTA) methods. In the current manuscript, we focused on NN-based optimization proxies as they are typically the first approach one would try before exploring more complex methods, making them a suitable baseline for comparison. Our primary objective was to introduce our new surrogate modeling approach and highlight its potential benefits. However, we agree that a broader comparison would strengthen the paper’s impact. To address this, we are willing to extend our computational results to include a Lagrangean dual-based NN surrogate, a competitive SOTA method that was initially proposed by Fioretto, Mak, and van Hentencryck in 2020. We would appreciate the reviewer’s guidance on whether this addition would be satisfactory or if they would suggest evaluating another method.
>
> 2. The motivation for using the hybrid vehicle control problem is based on its prior use in related works, such as Bertsimas and Stellato (2021), to demonstrate the utility of similar approaches. This allows for a direct comparison of our method’s performance within an established context. Additionally, the production scheduling case study is a classical textbook problem in production and manufacturing domains. Its continuous-time representation presents more challenging instances, enabling us to showcase the robustness and effectiveness of our approach in solving complex, real-world problems. As such, although our case studies do not use real-world data, we would argue that they are still realistic.
> We are happy to augment the results for the second case study by including instances that are significantly harder to solve. These instances were not included in the original manuscript because they could not be solved within a few hours, resulting in suboptimal solutions. However, we will incorporate these more challenging instances to further demonstrate the capabilities of our approach.
>
> 3. We thank the reviewer for highlighting this important point. We agree that the iterative nature of our penalty-based BCD algorithm, which requires solving optimization problems in each iteration, can become computationally prohibitive for large-scale instances. During the initial phase of the project, we explored alternative methods but ultimately converged on the presented solution, which performed well in our computational experiments. Since the primary objective of this work was to introduce a new surrogate modeling approach and demonstrate its potential benefits, we focused on providing a working solution rather than a systematic comparison of different algorithms. Nevertheless, we acknowledge the importance of developing more efficient solution methods for larger, higher-dimensional problems and plan to address this in our future work.
> It is true that the feasibility restoration step solves an MILP; however, this MILP turns out to solve very easily since it only aims to find the closest integer-feasible point to a point that is already very close. We are happy to include some additional statistics in the paper showing the computational performance of this step.

---

> > ### Comment · Reviewer_RhjW · 2025-02-13
> >
> > I would like to thank the authors for their detailed responses to my review comments. I have reviewed the authors' responses to all reviewers and the updated submission. I appreciate their further explanations regarding the applicability of the proposed method in online settings and their detailed analysis of cases where the method did not perform well, which enhances the clarity of their work. Additionally, the comparisons with their selected state-of-the-art (Fioretto, Mak, and van Hentencryck, 2020) strengthen the paper.
> >
> > Overall, I appreciate the effort made to address the concerns raised.

---

### Decision · Action_Editor_Mw5A · 2025-03-11

**Recommendation:** Accept as is

**Comment:**

The paper presents a method to learn and compute surrogate LPs for MILPs that take less time to solve, yet still has optimum close to that of the original MILP. Specific attention is given to the goal of achieving feasibility of the surrogate optimum in the original MILP. This is a problem of broad interest in discrete optimization, while the problem itself is a machine learning problem requiring development of new/better ML techniques. After the author-reviewer discussion and paper revisions, reviewers are satisfied with the presentation and experimental evaluation of the proposed method. I recommend acceptance of this paper to TMLR, given that both the "Claims and Evidence" and "Audience" criteria are satisfied.

**Audience:**

Yes, all reviewers (and I) find the subject of study interesting, and it is clear that many others in the community will be interested in the findings.

**Claims And Evidence:**

Reviewers agree that, with the additional baseline of GNN-based optimization proxy, the paper now presents a convincing experimental case for their proposed method.

---

> ### Author Response · Authors · 2025-03-19
> **Camera ready version.**
>
> We have uploaded the camera-ready version and would like to express our sincere gratitude to the reviewers, action editor, and editors-in-chief for their valuable contributions and support.